# P300 Source Localization Contrasts in Body-Focused Repetitive Behaviors and Tic Disorders

**DOI:** 10.3390/brainsci7070076

**Published:** 2017-07-01

**Authors:** Geneviève Sauvé, Simon Morand-Beaulieu, Kieron P. O’Connor, Pierre J. Blanchet, Marc E. Lavoie

**Affiliations:** 1Department of Psychiatry, McGill University, Montréal, QC H3A 1A1, Canada; genevieve.sauve@mail.mcgill.ca; 2Cognitive and Social Psychophysiology Lab, Centre de Recherche de L’Institut Universitaire en Santé Mentale de Montréal, 7331 Hochelaga Street, Montréal, QC H1N 3V2, Canada; 3Department of Neurosciences, Université de Montréal, Montréal, QC H3T 1J4, Canada; simon.morand-beaulieu@umontreal.ca (S.M.-B.); pierre.j.blanchet@umontreal.ca (P.J.B.); 4Department of Psychiatry, Université de Montréal, Montréal, QC H3T 1J4, Canada; kieron.oconnor@umontreal.ca; 5Centre D’études sur les Troubles Obsessionnels-Compulsifs et les Tics, Centre de Recherche de L’Institut Universitaire en Santé Mentale de Montréal, 7331 Hochelaga Street, Montréal, QC H1N 3V2, Canada; 6Department of Stomatology, Faculty of Dental Medicine, Université de Montréal, Montréal, QC H3T 1J4, Canada

**Keywords:** body-focused repetitive behaviors, Tourette syndrome, tics, event-related potentials, electrophysiology, P300, context updating

## Abstract

Tic disorders (TD) and body-focused repetitive behaviors (BFRB) have similar phenotypes that can be challenging to distinguish in clinical settings. Both disorders show high rates of comorbid psychiatric conditions, dysfunctional basal ganglia activity, atypical cortical functioning in the prefrontal and motor cortical regions, and cognitive deficits. Clinicians frequently confound the two disorders and it is important to find reliable objective methods to discriminate TD and BFRB. Neuropsychological tests and event-related potential (ERP) studies have yielded inconsistent results regarding a possible context updating deficit in TD and BFRB patients. However, most previous studies did not control for the presence of comorbid psychiatric condition and medication status, which might have confounded the findings reported to date. Hence, we aimed to investigate the psychophysiology of working memory using ERP in carefully screened TD and BFRB patients excluding those with psychiatric comorbidity and those taking psychoactive medication. The current study compared 12 TD patients, 12 BRFB patients, and 15 healthy control participants using a motor oddball task (button press). The P300 component was analyzed as an index of working memory functioning. Results showed that BFRB patients had decreased P300 oddball effect amplitudes over the right hemisphere compared to the TD and control groups. Clinical groups presented different scalp distributions compared to controls, which could represent a potential endophenotype candidate of BFRB and TD.

## 1. Introduction

Tic disorders (TD) and body-focused repetitive behaviors (BFRB) are two motor-related neuropsychiatric afflictions that can be difficult to distinguish in clinical settings because they often co-occur and because their symptoms can easily be confounded [1,2]. Indeed, large proportions of TD patients present comorbid disorders such as BFRB, but also depression, anxiety, and obsessive-compulsive disorder (OCD) [3,4,5]. Furthermore, the elaborate movements involved in BFRB can resemble the complex motor tics found in TD [6]. Hence, finding a reliable method to distinguish between the two groups is important in order to develop more efficient and specific treatments.

The phenomenology of tics and BFRB considerably overlap. On the one hand, tics are repetitive and sudden movements or vocalizations that can range from simple actions (e.g., eye blinks) to complex sequences of movements (e.g., head and finger contortions) [7]. Tics can also take the form of self-inflicted repetitive actions such as head banging or teeth grinding. On the other hand, BFRB is an umbrella term for debilitating, repetitive habits and behaviors that target one or more body regions. BFRB encompasses non-functional and distressing habits such as hair-pulling, skin-picking, nail or lip-biting [8]. The symptoms found in different types of BFRB are heterogeneous. However, they all represent movements oriented toward the body, and occur in reaction to feelings of discomfort and such feelings can also trigger tics [9]. Thus, TD and BFRB patients report an urge or impulse to perform the motor act (i.e., the tic or the repetitive behavior) and both experience relief after it has occurred [10,11,12,13]. In both conditions, chronic symptoms wax and wane, follow the same loop of negative reinforcement, and can be accentuated by stress, fatigue, or boredom [1,14,15]. To our knowledge, no study has reported the comorbidity rates between tic disorders and BFRB as a whole. Yet, few studies reported such rates for individual BFRB. For instance, trichotillomania was reported in about 3% of TD patients [4,16], but could occur more frequently in TD+OCD patients [17]. In a sample of 35 TD patients, Ghanizadeh and Mosallaei [18] reported nail-biting in almost 30% of them. This comorbidity between TD and BFRB highlights the need for objective assessment tools to distinguish them.

In addition to a similar symptomatology, both disorders share biological and neurocognitive characteristics. The current hypothesis is that TD and BFRB patients show defects of the basal ganglia [19,20,21,22,23,24]—a deep cerebral structure involved in the production and inhibition of movement [25]—and abnormal cortical activity in prefrontal and motor cortical regions [21,26,27,28,29]. Neurocognitive impairments have been reported for TD and BFRB patients in multiple domains, such as attention [30,31,32], inhibition of motor responses [14,33,34,35,36,37], and spatial working memory [30,38,39,40,41]. Findings regarding working memory seem less consistent, as some studies have reported no such deficit [42,43,44,45]. However, these studies rarely exclude patients presenting comorbid disorders or taking medications, which may explain the heterogeneity of results found in working memory tasks.

The investigation of cognitive deficits in medicated patients suffering exclusively from TD (i.e., without comorbid attention deficit hyperactivity disorder (ADHD), OCD, depression, anxiety, or psychosis) showed deficits in behavioral inhibition (i.e., sentence completion), but not in memory and executive functioning [42]. These neuropsychological tests were probably not sensitive enough to detect cerebral impairments. Such a situation has been observed for TD patients, where cortical hyperactivity over the parietal and frontal regions were found during a working memory task, using functional magnetic resonance imaging (fMRI), while no deficits were identified using neuropsychological testing [46]. This could suggest that, to a certain extent, patients with TD and BFRB may present cerebral compensatory mechanisms allowing neuropsychological performances comparable to control participants, despite frontal and parietal impairment [47]. Hence, other effective techniques, such as fMRI, electroencephalography (EEG), and event-related potentials (ERP), could help to distinguish between the two disorders and pinpoint neurocognitive differences between them.

Both disorders show problems in planning action and adapting action to achieve goals. In the current study, a group of TD was compared to a group of BFRB during a motor oddball task and the recording of the P300 component, which indexes context updating processes. Only a handful of studies have applied this technique in groups of TD patients, and even less so in BFRB patients. O’Connor et al. [48] reported a near absence of synchrony between ERP activities reflecting motor preparation and motor execution in 13 medicated TD patients and 17 patients with BFRB. On the one hand, Roberts et al. [49] showed a decreased error-related negativity, associated with less self-monitoring in 10 undiagnosed individuals suffering from symptoms of BFRB. On the other hand, Johannes et al. [50] found an increased error-related negativity in 10 TD patients taking psychoactive medication and not screened for clinical comorbidity. Few studies in TD patients [50,51,52,53,54,55,56,57] reported additional findings regarding the P300 component, with some inconsistency between studies. Johannes et al. [51] first reported a reduced P300 in 12 TD patients, during a motor oddball task. Later, the same research group reported no significant difference in the P300 amplitude of 10 TD patients, compared to 10 controls during another type of motor oddball task [50]. However, these patients were under medication and presented clinically significant levels of comorbid conditions (e.g., OCD, ADHD), which could explain the discrepancy in findings. In this vein, reduced P300 amplitude in TD+ADHD patients was found during a Go/NoGo task, while TD-only patients had normal P300 amplitude [57]. Similar result was found in a stimulus-response mapping task, where a trend toward a P300 amplitude decrease was found in TD+ADHD but not TD-only patients [56]. In a flanker task, Eichele et al. [55] reported higher amplitude of the early P300 (P3a) in TD patients, but no significant difference regarding later positive components.

The oddball task aims to alter the preprogramed execution of a response, by introducing rare stimuli (that have a lower probability of occurrence) into a sequence of frequent stimuli. This task elicits a P300, which indexes mental representations updating in working memory [58]. When a rare stimulus appears, it indicates that a different type of response must be executed, which activates the prefrontal cortex in order to inhibit that preprogramed response [59]. As noted, this could be a dysfunction in TD and BFRB, whom have already shown problems in countermanding actions, when plans need to be inhibited and reset.

As mentioned earlier, the inclusion of patients under medication could have confounded earlier results. Furthermore, many studies also adopted a clinically ecological approach in which they included TD and BFRB patients with other psychiatric comorbidities. These comorbidities, especially OCD and ADHD, have been shown to enhance neurocognitive deficits in TD [60]. Therefore, the current study compared TD and BFRB patients free of medication and comorbid symptoms to investigate the chronometry of working memory processes using a motor oddball task. By directly comparing non-comorbid and unmedicated TD and BFRB patients, psychophysiological markers would be more specific and allow more precise clinical differentiation. As treatments for both TD and BFRB are currently being developed [2], identifying specific methods to distinguish patients from both populations also has the potential to enhance treatment efficacy. Based on the previous ERP literature in TD and BFRB, we hypothesize that both patient groups will show reduced P300 amplitude compared to controls, and that TD should show more prominent P300 scalp topography over motor regions, while BFRB should activate a more widespread activation of the same component [2,51,52,53].

## 2. Methods

### 2.1. Participants

Patients with either TD or BFRB were initially recruited to take part in a wider program investigating the efficacy of a novel cognitive-behavioral therapy [61,62]. In the present study, 24 patients (12 with TD and 12 with BFRB) were carefully screened to exclude those with comorbid condition(s) and those taking psychoactive medication. Nine of the 12 TD patients and 10 of the 12 BFRB patients were also included in a previous study using a different experimental task [2]. Information on the therapy program and inclusion and exclusion criteria for the therapy can be found in previous publications [63,64]. Patients with the following BFRB subtypes were included in the BFRB group: hair-pulling (trichotillomania, *n* = 6), skin-picking (excoriation, *n* = 1), nail-biting (onychophagia, *n* = 4), and bruxism (*n* = 1). Initially, a clinical evaluator assessed the symptoms (tics and BFRB) to ensure the diagnosis of all patients. Subsequently, participants received a clinical psychological assessment (supervised by K.P.O.) and then met with a neurologist (P.J.B.) to screen for neurological disorders. The specific exclusion criteria for the current study were: (i) the presence of any treatment (pharmacological or other) for tics or BFRB, (ii) any history of DSM-IV-TR Axis I disorder other than Tourette syndrome, persistent tic disorder or BFRB, (iii) presence of personality disorders (assessed by PDQ-4 scale [65]), (iv) dependence on alcohol or drugs. A group of healthy control participants (*n* = 15) was recruited via local media to match patients on age, sex, and non-verbal intelligence (evaluated with Raven’s matrices [66]). Control participants were screened for psychiatric or neurological diseases and were not taking psychoactive medication. All data presented here were collected prior to the beginning of the therapy for which tic and BFRB patients were initially recruited [63].

The Tourette Syndrome Global Scale [TSGS; 68] and the Yale Global Tic Severity Scale [YGTSS; 69] were used to uniformly quantify tics and BFRB. The TSGS combines a clinical evaluation with reports made by patients and parents to assess tics (simple/complex and motor/phonic) and social functioning in the past week. Global scores vary from 0 (no symptoms) to 100 (worst Tourette possible). Inter-rater reliability, construct validity, and concurrent validity were all found to be good [67]. The YGTSS is a symptom checklist for tic severity (number of tics, frequency, intensity, complexity and interference) and functioning impairments (distress experienced in interpersonal, academic and occupational spheres) over the previous week. The YGTSS global score (0–100) is the sum of subscales evaluating the severity of tics (0–50) and daily impairment (0–50). The YGTSS has demonstrated good internal consistency, test-retest stability, inter-rater agreement and convergent validity (with the TSGS) [68,69,70].

Both the TSGS and the YGTSS were used “as is” in the TD group and adapted to assess the presence of habits in the BFRB group. In these adapted versions, the word “tic” was replaced by the word “habit”. The correlations between the adapted scales and appropriate instruments measuring BFRB frequency and severity were reported to be good [1,61]. The Massachusetts General Hospital Hair Pulling Scale (MGH-HPS; [71]) was additionally administered to assess BFRB severity. The MGH-HPS is a seven-point inventory measuring the severity of trichotillomania symptoms. Again, an adaptation of this scale was proposed to assess onychophagia, skin picking, and skin scratching. This scale was thus used to quantify the severity of the principal BFRB of each patient.

Since the focus was to investigate patients unaffected by comorbid disorders, a thorough examination of the most frequent comorbid afflictions was made using the SCID-I/P [72]. Levels of anxiety (Beck Anxiety Inventory (BAI); [73]), depression (Beck Depression Inventory (Beck Depression Inventory (BDI); [74]), and obsessive-compulsive (Vancouver Obsessional Compulsive Inventory (VOCI); [75]) were also assessed (Table 1). All participants were right-handed (Edinburgh Handedness Inventory (Edinburgh Handedness Inventory; [76]), had normal or corrected to normal visual acuity (Snellen notation system) and gave their written informed consent before participating. The study was approved by the local ethic committee of the *Institut universitaire en santé mentale de Montréal*.

### 2.2. Experimental and Recording Settings

#### 2.2.1. Experimental Task: The Motor Oddball Paradigm

A classical oddball task was administered to all participants. Stimuli were presented using the Presentation software (Neurobehavioural Systems, Albany, CA, USA) and consisted of 200 black letters (Xs and Os) in Arial font (size = 48) on a white background displayed in a pseudo-random order. The letter “O” was the frequent stimulus (probability = 0.80) and the letter “X” was the rare stimulus (probability = 0.20). Stimuli were presented during 100 ms each at the center of a monitor screen (Viewsonic SVGA 17 inches flat screen monitor) with an inter-stimulus interval randomly ranging from 1700 to 2200 ms. Participants were seated in a dimly lit room with their head positioned 90 cm from the monitor. Before the beginning of the task, they were instructed to press the left arrow of the keyboard with their left hand when they perceived a rare (letter X) stimulus and the right arrow with their right hand for frequent (letter O) stimuli.

#### 2.2.2. Electrophysiological Recordings

The raw EEG was recorded with a digital amplifier (Sensorium, Inc., Charlotte, VT, USA) from 60 Ag-AgCl electrodes, mounted in a Lycra cap (Electrode Arrays, El Paso, TX, USA) and placed according to the extended international 10/20 system [77]. Thirty-six electrodes were selected for our specific statistical analyses and grouped in three regions and two hemispheres: Frontal (AF3, AF1, F5, F3, F1, AF4, AF2, F6, F4, F2), Central (FC3, FC1, C5, C3, C1, FC4, FC2, C6, C4, C2), and Parietal (CP5, CP1, P5, P3, P1, CP6, CP2, P6, P4, P2). The midline was also included in our analyses (AFZ, FZ, FCZ, CZ, CPZ, PZ). Electrodes were referenced to the nose and their impedances were kept below 5 KΩ. Recordings were continuously sampled at 500 Hz, amplified with a calibrated gain of ±10,000 and filtered between 0.01–100 Hz. Additionally, electro-oculograms (EOG), placed at the outer canthus of each eye and at infra- and supraorbital position of the right eye, were recorded from 4 bipolar electrodes for offline ocular corrections. Data were acquired by the IWave software (InstEP Systems Inc., Montréal, QC, Canada).

#### 2.2.3. ERP Extraction

Ocular artifacts that contaminated the EEG signals were corrected offline using the Gratton algorithm [78]. Subsequently, raw EEG signals were averaged offline, time-locked to the stimuli onset, in a time window of 100 ms prior to stimulus onset until 900 ms after stimulus onset. The averaging procedure was completed separately for the two conditions, corresponding to rare and frequent categories. Averaged signals were digitally filtered offline with low- and high-pass filters set at 0.3 Hz and 30 Hz, respectively. Remaining signals exceeding 100 μV and clippings due to amplifier saturation were eliminated during the averaging procedure. Epochs containing fewer than 20 trials for each category were eliminated. The P300 component was scored as the maximum peak during the 300–550 ms post-stimulus interval.

### 2.3. Statistical Analyses

Sociodemographic, clinical, and behavioral data were analyzed with one-way ANOVAs when a three groups comparison was involved, while independent-sample *t-*tests (bilateral α = 0.05) or Kruskall–Wallis tests (for non-parametric data) were used when the comparisons involved only the two clinical groups. Tukey post hoc tests were used when variances were homogeneous (according to Levene’s test), while Games-Howell tests were utilized for heterogeneous variances. The electrophysiological data were analyzed using repeated measures ANOVAs and the Greenhouse-Geisser correction was applied when the sphericity assumption was not met. One between-subject factor with three levels (Group: TD, BFRB, Control) was applied and 4 within-subject factors that comprise: Condition (Rare, frequent), Region (Frontal, central, parietal), Hemisphere (Left, right) and Electrode sites (6 levels as follow: Left Frontal = AF3, AF1, F5, F3, F1; Right Frontal = AF2, AF4, F2, F4, F6; Left Central = FC3, FC1, C5, C3, C1; Right Central = FC2, FC4, C2, C4, C6; Left Parietal = CP5, CP1, P5, P3, P1; Right Parietal = CP2, CP6, P2, P4, P6). Additionally, analyses were conducted with midline electrodes with the same between-subject factor and the following within-subject factors: Condition (Rare, frequent) and Electrode sites (6 levels; AFZ, FZ, FCZ, CZ, CPZ, PZ).

### 2.4. P300 Source Localization

In a subsequent analysis, neural generators of the averaged ERPs were analyzed with sLORETA software (University Hospital of Psychiatry, Zürich, Switzerland) in the P300 time interval (300–550 ms post-stimulus) showing significant differences between experimental conditions (rare vs. frequent). sLORETA uses a distributed source localization algorithm to solve the inverse problem of brain electric activity [79], regardless of the final number of neural generators [79,80]. The sLORETA algorithm calculates the current density values (unit: amperes per square meter; A/m^2^) of 6.239 gray matter voxels belonging to the brain compartment with a spatial resolution of 5 mm × 5 mm × 5 mm each. The whole three-dimensional brain compartment comprises cortical gray matter and the hippocampus only and does not contain any deep brain structures such as the thalamus or the cerebellum. Anatomical regions are labeled according to (1) the probabilistic MNI-152 template made digitally available by the Brain Imaging Center of the Montreal Neurological Institute [81] and (2) the Talairach Daemon [82]—a digitized version of the Co-Planar Stereotaxic Atlas of the Human Brain introduced by [83].

To display differences for the P300 time window between the two experimental conditions (e.g., “Rare deviant” > “frequent standard” for the oddball paradigm), statistical non-parametric mapping (SnPM) as introduced by Nichols and Holmes [84] was used to compute the averaged intracerebral current density distribution at time intervals showing significant differences based on non-parametric voxel-by-voxel one-tailored paired samples log of f-ratio of averages (with 5000 permutations) on the three-dimensional sLORETA images. Statistical significance was assessed by defining critical thresholds (critical *t*) corrected for multiple comparisons (*p* < 0.01 and *p* < 0.05, respectively) for all tested voxels. The same procedure was done to display differences to compare the three group differences on the P300 time oddball effect (e.g., “tic disorders” > “control”; “tic disorders” > “BFRB”; “BFRB” > “control”).

The null hypotheses equaled the assumption that there were no differences among experimental conditions. Current density values at each voxel were computed in the solution space as a linear and weighted sum of the scalp electric potentials. Activation of a given voxel was based on the smoothness assumption, meaning that neighboring voxels show a highly synchronous activity [85]. Support comes from electrophysiological studies showing that neighboring neural populations show a highly correlated electrical activity [85,86]. As proposed by Friston [87], activated voxels exceeding critical t values were considered as being regions of cortical activation. Finally, statistical analysis resulted in an averaged corresponding three-dimensional intracerebral current density distribution and obtained cortical regions were classified about their corresponding BA [88] and normalized coordinates (Talairach and MNI, respectively).

## 3. Results

### 3.1. Socio-Demographic and Clinical Description

No significant group difference was found regarding age, education level, and non-verbal intelligence (Table 1). Gender ratio was significantly different between TD and BFRB groups (H(2) = 6.55, *p* = 0.04) (followed by post hoc Mann–Whitney Test U = 36, *p* = 0.04, *r* = −0.52). All groups were comparable regarding depression levels, but BFRB patients showed significantly higher anxiety scores than the control group (F(2,36) = 3.92, *p* = 0.03, *r* = 0.30). Both clinical groups were comparable regarding obsessive-compulsive symptoms.

### 3.2. Reaction Times and Performance during the Oddball Task

A significant Condition main effect (F(1,38) = 134.17, *p* < 0.001, partial ŋ^2^ = 0.78) revealed faster responses in the frequent (421 ms) than in the rare (491 ms) condition. There was no group difference regarding reaction times. Additionally, all participants showed excellent performance accuracy in rare (93% average accuracy) and frequent (97% average accuracy) conditions, with no group difference at that level.

### 3.3. Amplitude of the P300 Component

The P300 component peaked at an average latency of 396 ms after stimulus onset. The omnibus ANOVA revealed significant Condition (F(1,36) = 149.71, *p* < 0.001, partial ŋ^2^ = 0.81), Region (F(2,72) = 27.25, *p* < 0.001, partial ŋ^2^ = 0.43), and Electrode Site (F(4,144) = 93.92, *p* < 0.001, partial ŋ^2^ = 0.72) main effects. Furthermore, a Condition by Hemisphere by Electrode by Group four-way interaction reached significance (F(8,144) = 3.16, *p* = 0.005, partial ŋ^2^ = 0.15). This four-way interaction was decomposed by condition. No group interaction was significant in the frequent condition. However, the Hemisphere by Electrode by Group (F(8,144) = 3.97, *p* < 0.001, partial ŋ^2^ = 0.18) and Region by Hemisphere by Electrode by Group (F(16,288) = 2.01, *p* = 0.04, partial ŋ^2^ = 0.10) interactions reached the significance threshold in the Rare condition, meaning that group differences were mainly found in the Rare condition.

The four-way interaction of Region by Hemisphere by Electrode by Group was decomposed by hemisphere. There was no significant group interaction within the left hemisphere. A significant Electrode by Group (F(8,144) = 2.73, *p* = 0.02, partial ŋ^2^ = 0.13) interaction was found in the right hemisphere, which indicates between-group differences over the right hemisphere, in response to the rare condition. The latter interaction of Electrode by Group was decomposed according to the following electrode lines (i.e., 1st AF4/FC4/CP6; 2nd AF2/FC2/CP2; 3rd F6/C6/P6; 4th F4/C4/P4; 5th F2/C2/P2). BFRB patients showed a significantly reduced P300 amplitude compared to controls, over the 1st line AF4/FC4/CP6 (F(2,36) = 5.49, *p* = 0.008, partial ŋ^2^ = 0.23; Bonferroni: *p* < 0.01), 3rd line F6/C61P6 (F(2,36) = 7.77, *p* = 0.002, partial ŋ^2^ = 0.30; Bonferroni: *p* < 0.005), 4th line F4/C4/P4 (F(2,36) = 4.44, *p* = 0.02, partial ŋ^2^ = 0.20; Bonferroni: *p* < 0.05), and 5th line F2/C2/P2 (F(2,36) = 4.17, *p* = 0.02, partial ŋ^2^ = 0.19; Bonferroni: *p* < 0.05) electrode lines, as revealed by significant Group main effects (Figure 1). Furthermore, BFRB patients also showed a significantly reduced P300 compared to TD patients over the 3rd electrode line F6/C6/P6 (*p* = 0.02; Bonferroni: *p* < 0.05). Despite a general tendency toward a P300 reduction, TD patients showed amplitudes that were statistically comparable to control participants. The aforementioned electrophysiological differences between groups were not explained by BFRB patients’ higher scores on the BAI since the Condition by Hemisphere by Electrode by Group interaction from the omnibus test remained significant after covarying for BAI scores (F(8,140) = 2.33, *p* = 0.03).

#### 3.3.1. sLORETA Source Localization—The Rare–Frequent Oddball Contrast

The statistical comparison between experimental conditions “frequent” and “rare” in the peak activation within a time window between 300 and 550 ms (corresponding to the P300 oddball effect) was done across the three groups separately. Results of Table 2 showed that in the TD group, a significantly (*p* between 0.0012 and 0.0018) larger cortical activation of the oddball effect at 518 ms post-stimulus was expressed in the superior and middle frontal gyri which corresponds to BA6 and BA10 (Figure 2). The Table 3 showed that in the BFRB group a larger oddball effect activation at 402 ms post-stimulus was significant (*p* between 0.0014 and 0.0001) in the right parietal lobe, mainly within the precentral (BA43) and postcentral (BAs 1/2/3) gyri as well as the right inferior parietal gyrus (BA 40). Other right hemisphere structures are also involved, including mainly the right superior temporal gyrus (BAs 42/22), the inferior frontal gyrus (BAs 9/44/45), and the frontal precentral gyrus (BA 4/6) (Figure 3). The Table 4 showed that in the control group (Figure 4) a larger oddball effect activation at 348 ms post-stimulus is significant (*p* between 0.0010 and 0.0022) over the right occipital region of the cuneus (BA18) and precuneus (BAs 7/19/31), as well as activation over the parietal postcentral gyrus (BA3).

#### 3.3.2. The Group Comparison Contrast

The statistical comparison between groups (“control”; “tic” and “BFRB”) in the peak activation within a time window between 300 and 550 ms was done on the rare–frequent subtraction. Results of Table 5 showed that in the control/BRFB comparison, a significantly larger difference in cortical activation of the oddball effect peaked at 366 ms post-stimulus significant (*p* between 0.006 and 0.012) in the right hemisphere superior frontal gyrus (BA6) and middle temporal gyrus (BA39) (Figure 5) confirming that BFRB patients showed a significantly reduced P300 oddball effect compared to the controls, mainly over the right superior frontal gyrus. Table 6 showed that, in the tic/BFRB comparison, a significantly larger difference in cortical activation of the oddball effect peaked at 324 ms post-stimulus significant (*p* between 0.0004 and 0.0040) in the right hemisphere middle temporal gyrus (BA22) and middle temporal gyrus (BA39) (see Figure 6) confirming that BFRB patients showed a significantly reduced P300 oddball effect compared to the TD patients, mainly over the right middle temporal gyrus. In the control/tic group comparison (Figure 7) a larger difference in cortical activation of the oddball effect peaked at 356 ms post-stimulus, but there is no significant difference between groups (*p* between 0.1708 and 0.2942).

## 4. Discussion

The current study aimed at investigating P300 processes during a motor oddball task, in carefully selected unmedicated patients affected solely by TD or BFRB, without comorbid diagnosis. We initially hypothesized that both patient groups would show reduced P300 amplitude compared to controls, and that TD should have more prominent P300 scalp topography over motor regions, while BFRB should activate a more widespread activation of the same component suggesting qualitatively different patterns of electrocortical activity related to P300 context updating process. ERP analysis revealed that patients with TD showed no significant P300 effect, while the BFRB group presented a significant P300 amplitude reduction, shifted over the right temporal region, in comparison to both TD and control groups. sLORETA analysis further confirmed that the two clinical groups exhibited different scalp distributions of the oddball effect, compared to the control group. The contrasts between groups also revealed significant differences between the control and BFRB, and between BFRB and TD, but no current source topography differences between the control and the TD group.

The P300 is considered to be influenced, in part, by context updating in working memory [90], as well as stimulus evaluation and categorization [91,92]. The P300 amplitude also indexes processing related to the maintenance of the surrounding environment’s exact representation in the brain [93]. It has been proposed that the P300 component would index brain activities underlying revision of the mental representation induced by incoming stimuli [94,95]. Concerning the scalp topography, the P300 has previously shown a robust temporo-posterior distribution in healthy subjects. By combining ERPs and fMRI, Li et al. [96] observed that the P300 was generated in the parietal lobules, frontal and central gyri, and in the anterior cingulate cortex. Earlier research involving healthy participants with the motor oddball task showed localized activation of the supplementary motor area, the cerebellum, the thalamus, and the parietal cortex. An additional activation of the middle frontal gyrus was observed, along with the central opercular cortex and the parietal operculum, which reflect integration of sensory and motor responses [97]. Some of these structures are affected in BFRB patients [21], which show increased gray matter density in the cingulate and the parietal regions. Other results also proposed, that TD patients show compensatory activity, notably within the prefrontal and cingulate cortical regions [98]. Our results showed that while all groups presented similar RTs and response accuracy, only patients with BFRB had a significantly smaller P300 oddball effect compared to the other two groups. In addition, the source localization of the oddball effect clearly reveals a right hemisphere shift within the inferior parietal lobule, the postcentral and the superior temporal gyri. Hence, the increased gray matter density in the cingulate and parietal regions, previously reported in BFRB patients, might underlie an increased mobilization of available resources to perform that task at the same level as the controls. Correspondingly, the P300 oddball effect decrease in BFRB patients could reflect a failure to shift their attentional focus toward the actual context, since they are coping with thoughts patterns and feelings driving them to perform their BFRB [99]. This could partially explain the difference in P300 amplitude and the specific right hemisphere scalp topography between BFRB patients and healthy controls. A recent review underlined the prominent role for the right hemisphere—particularly the right inferior parietal lobule—involved in representing regularities and in distinguishing when new information is congruent or discrepant with respect to the predictions generated by a currently held mental model [100]. Another way to approach that question is to integrate a cognitive, behavioral and physiological scheme to functionally link these symptom clusters with neurocognitive processes. For instance, according to the cognitive-behavioral and psychophysiological model [99], context updating processes are important in both TD and BFRB populations. Within that perspective, patients have a specific style of action planning, which drives them to update the production and reinforcement of their tics and BFRB continuously (i.e., the context). If we assume that the right inferior parietal lobule is involved in the processing of salient and surprising information with respect to a current internal model, this mean that the BFRB would overactivate the right hemisphere regions (including the inferior parietal lobule) in order to reinforce new information that conforms to model-generated expectations [100]. Indeed, these patients have the perfectionist desire to always stay ahead of themselves. Therefore, this inclination leads to set their attentional focus toward unrealistic goals, and not toward the “here and now” or the actual model-relevant context [99].

Interestingly, recent studies reported enhanced frontal P300 amplitudes in unmedicated OCD patients [101,102,103,104,105,106,107], which suggests that BFRB and OCD patients may be distinct disorders with similar behavioral symptoms, while showing dissimilar P300 pattern. During a countermanding motor task, TD and BFRB patients both failed to adjust their hand responses to automated or controlled movements [48]. To perform that motor task, TD patients had the most severe divergence in the synchrony between their electrocortical activity (readiness potentials) and their actual motor responses, followed by the BFRB and the control group. Their findings gave support to a dimensional model of classification with BFRB falling closer to TD than to the control group along a continuum of motor arousal. Altogether, these results further showed that BFRB, TD, and OCD correspond to a clinical spectrum that could share some commonalities and discrepancies in their cerebral motor networks.

Earlier findings suggest that comorbidity or medication status may have influenced the interpretation of previously reported ERP recordings of the P300 that often showed a reduction in amplitude in TD [51,108]. Therefore, the presence of comorbid psychiatric disorders as well as adrenergic medications may both impact on the P300, which would lead to atypical electrocortical activity, not entirely related to tic disorders. The presence of attention deficit hyperactive disorder—a frequently comorbid disorder with TD—is also associated with irregular release of norepinephrine in frontal areas [109], which could also impact on the P300 amplitude. To that extent, pharmacological treatment for tics, such as Clonidine and Guanfacine (noradrenergic receptor antagonists [110]), may have involved the cingulate cortex, a region previously identified as one of multiple P300 generators [96,111]. Given that norepinephrine is released in the cingulate cortex among other regions, the reduction in P300 amplitude may then be related to medication status rather than TD per se. Also, we cannot rule out the possibility of an overlap of movement-related potentials with the P300 component [112], which could have affected P300 amplitude in TD patients. This is especially true since increased lateralized readiness potential amplitude has been reported in TD patients [113].

## 5. Conclusions

To sum up, our study showed decreased P300 amplitudes and different distribution of cortical activity in unmedicated patients solely affected by BFRB (i.e., without comorbid psychiatric condition). This pattern of electrocortical activation represents a potential endophenotype candidate of BFRB where the P300 amplitude scalp topography may reveal important clues of subtle alterations, probably in the gray matter density of the cingulate and parietal regions. While our study has some limitations, as the small sample size, the exclusion of medicated and comorbid TS patients represents a strength. In this vein, future studies are encouraged to control for medication and comorbidities such as depression, anxiety or obsessive-compulsive disorders given their important influence on ERP recordings. Moreover, our results support previous findings of an unaltered P300, measured by a motor oddball task, in unmedicated patients with TD as their sole psychiatric affliction [50,52]. Future research could include patients suffering from obsessive-compulsive disorders, when comparing ERP patterns of unmedicated patients suffering from TD and BFRB, given the high comorbidity between those three disorders. According to the DSM-5 [114], trichotillomania and skin picking are classified in the obsessive-compulsive and related disorders category, while onychophagia and dermatophagia are mentioned as “other specified obsessive-compulsive and related disorders”. Although they are all classified in the same category and broadly conceptualized as BFRB, their clinical presentation can be rather heterogeneous, which could confound our results. Yet, given the recent classification of these disorders, in the category of obsessive-compulsive and related symptoms, the inclusion of a group of OCD patients might deepen our understanding of the neurocognitive chronometry specific to BFRB.

## 6. Clinical Implications

Unmedicated TD and BFRB patients with no psychiatric comorbidity showed distinct P300 scalp topography patterns during a motor oddball task. This could constitute a promising electrocortical marker distinguishing BFRB from TD.

## 7. Limitations

Future studies could include a group of patients afflicted with obsessive-compulsive disorder given their high comorbitity rates with TD and BFRB to further refine distinctions. These results require replications with larger sample size.

## Figures and Tables

**Figure 1 brainsci-07-00076-f001:**
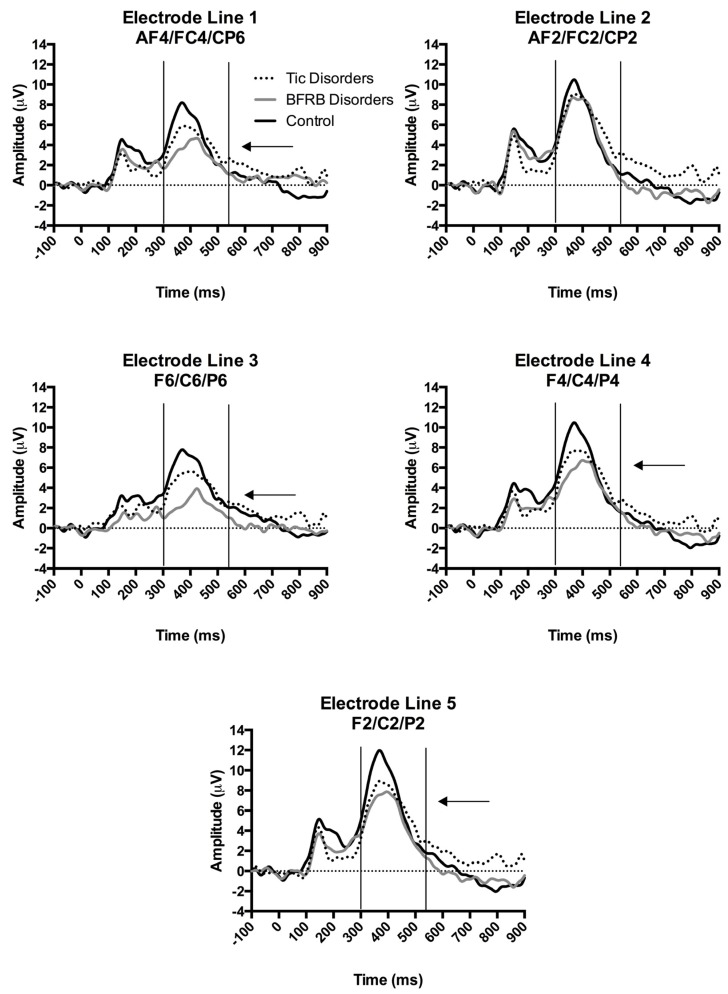
ERPs in the rare condition over the right hemisphere according to a division by electrode line. The x-axis represents the amplitude (μV) and the y-axis represents the time in milliseconds. The ERPs were time-locked to the stimulus onset. The P300 was measured from baseline-to-peak between 300–550 ms post-stimulus onset. Arrows show significantly smaller P300 voltages for BFRB patients compared to control participants over the first, third, fourth and fifth electrode lines.

**Figure 2 brainsci-07-00076-f002:**
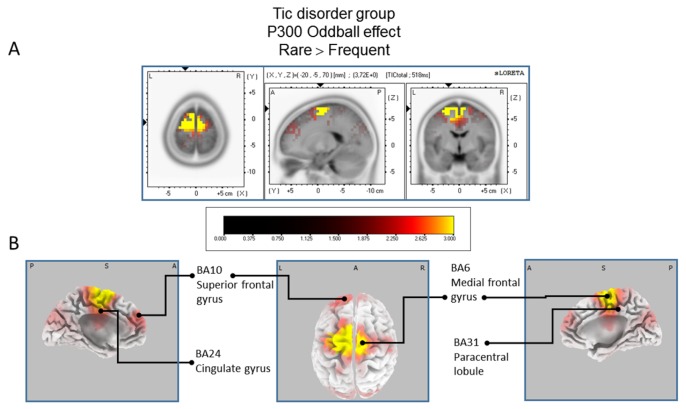
Results of the tic disorder patients based on the standardized low-resolution brain electrotomography (sLORETA) current source density localization of the oddball effect (contrast: “Rare” > ’’Frequent”) in the peak activation of the P300 component within a 300–550 ms time window. Images have been obtained after statistical non-parametric mapping (SnPM) and co-registration to the stereotaxic Talairach space based on the Co-Planar Stereotaxic Atlas of the Human Brain [83] and the probabilistic MNI-152 template [81]. Activated voxels are indicated by yellowish and reddish colors (after correction for multiple comparisons (*p* < 0.01 and *p* < 0.05, respectively)). (**A**) The peak of highest cortical activity has been found in parts of the superior frontal gyrus (BA10), the medial frontal gyrus (BA6), the paracentral lobule (BA31), and the cingulate gyrus (BA24). (**B**) A shifted transverse and sagittal view of the left and the right hemisphere, showing cortical activations on the three-dimensionally rendered Colin27 template [89] L, left; R, right; A, anterior; P, posterior; MNI, Montreal Neurological Institute; X, Y, Z, corresponding MNI coordinates; BA, Brodmann Area.

**Figure 3 brainsci-07-00076-f003:**
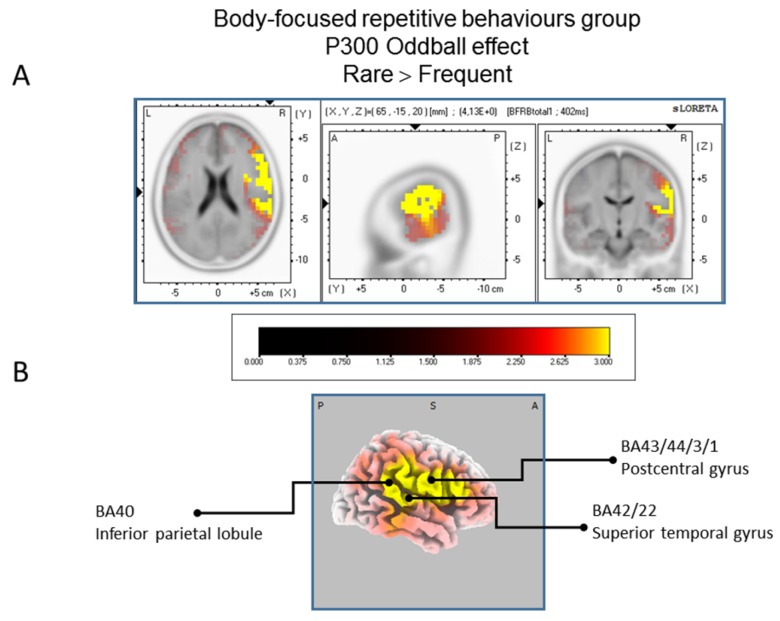
Results of the Body-focused Repetitive Behaviors patients based on the standardized low-resolution brain electrotomography (sLORETA) current source density localization of the oddball effect (contrast: “Rare”> Frequent”) in the peak activation of the P300 component within a 300–550 ms time window. Images have been obtained after statistical non-parametric mapping (SnPM) and co-registration to the stereotaxic Talairach space based on the Co-Planar Stereotaxic Atlas of the Human Brain [83] and the probabilistic MNI-152 template [81] Activated voxels are indicated by yellowish and reddish colors (after correction for multiple comparisons (*p* < 0.01 and *p* < 0.05, respectively)). (**A**) The peak of highest cortical activity has been found in parts of the inferior parietal lobule (BA40), the postcentral gyrus (BA43/3/1) and the superior temporal gyrus (B42/22). (**B**) A shifted lateral view of the right hemisphere, showing cortical activations on the three-dimensionally rendered Colin27 template [89]. A, anterior; P, posterior; MNI, Montreal Neurological Institute; X, Y, Z, corresponding MNI coordinates; BA, Brodmann area.

**Figure 4 brainsci-07-00076-f004:**
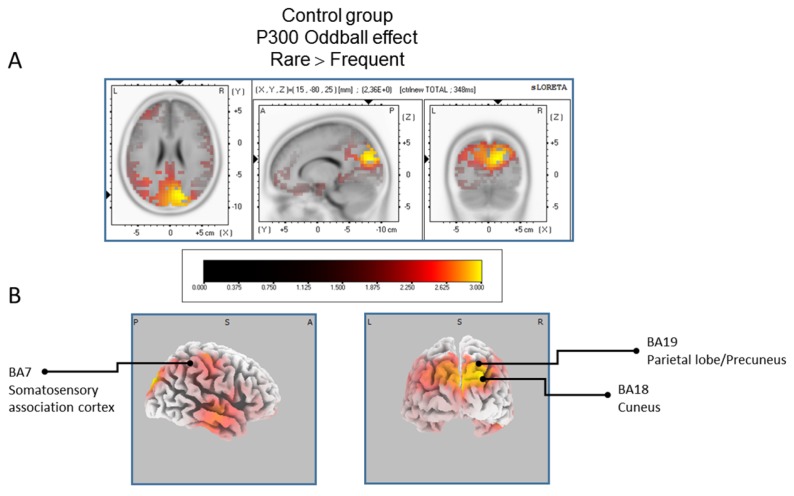
Results of the controls participant based on the standardized low-resolution brain electrotomography (sLORETA) current source density localization of the oddball effect (contrast: “Rare” > Frequent”) in the peak activation of the P300 component within a 300–550 ms time window. Images have been obtained after statistical non-parametric mapping (SnPM) and co-registration to the stereotaxic Talairach space based on the Co-Planar Stereotaxic Atlas of the Human Brain [83] and the probabilistic MNI-152 template [81]. Activated voxels are indicated by yellowish and reddish colors (after correction for multiple comparisons (*p* < 0.01 and *p* < 0.05, respectively)). (**A**) The peak of highest cortical activity has been found in parts of the cuneus (BA7/18) and the precuneus (BA19). (**B**) A shifted view of the right hemisphere and the posterior cortex, showing cortical activations on the three-dimensionally rendered Colin27 template [89]. L, left; R, right; A, anterior; P, posterior; MNI, Montreal Neurological Institute; X, Y, Z, corresponding MNI coordinates; BA, Brodmann area.

**Figure 5 brainsci-07-00076-f005:**
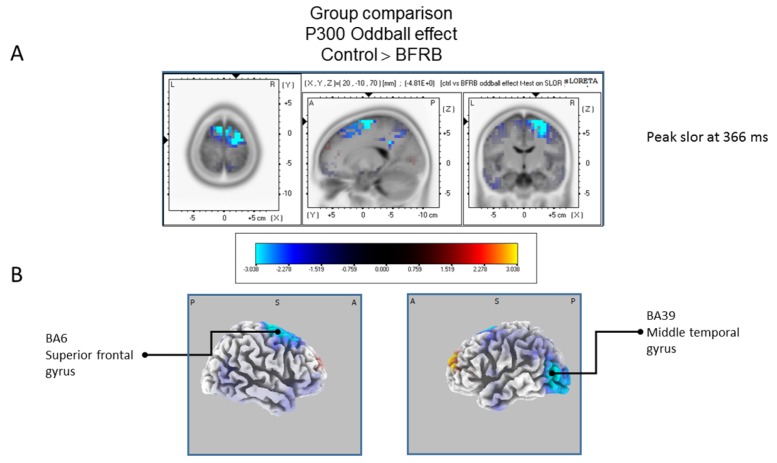
Comparison of the oddball effect between Control participants and BFRB patients based on the standardized low-resolution brain electrotomography (sLORETA) current source density localization of the group effect (contrast: “Ctrl” > BFRB”) in the peak activation of the P300 component within a 300–550 ms time windowImages have been obtained after statistical non-parametric mapping (SnPM) and co-registration to the stereotaxic Talairach space based on the Co-Planar Stereotaxic Atlas of the Human Brain [83] and the probabilistic MNI-152 template [81]. Activated voxels are indicated by blue colors [after correction for multiple comparisons (*p* < 0.05). (**A**) The peak of highest cortical activity group difference has been found in parts of the superior frontal gyrus (BA6) and the middle temporal gyrus (BA39). (**B**) A shifted transverse view of the left and the right hemisphere, showing cortical activations on the three-dimensionally rendered Colin27 template [89]. A, anterior; P, posterior; MNI, Montreal Neurological Institute; X, Y, Z, corresponding MNI coordinates; BA, Brodmann area.

**Figure 6 brainsci-07-00076-f006:**
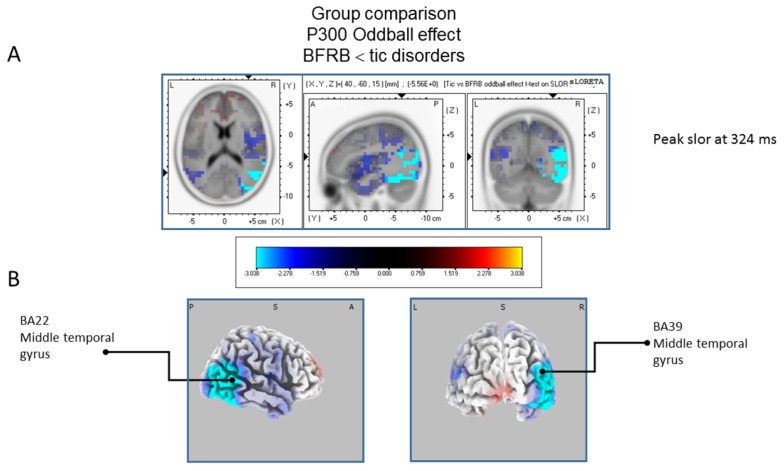
Comparison of the oddball effect between Tic and BFRB patients based on the standardized low-resolution brain electrotomography (sLORETA) current source density localization ofthe group effect (contrast: “BFRB’’ < ‘‘tic”) in the peak activation of the P300 component within a 300–550 ms time window. Images have been obtained after statistical non-parametric mapping (SnPM) and co-registration to the stereotaxic Talairach space based on the Co-Planar Stereotaxic Atlas of the Human Brain [83] and the probabilistic MNI-152 template [81]. Activated voxels are indicated by blue colors (after correction for multiple comparisons (*p* < 0.05)). (**A**) The peak of highest cortical activity of group difference has been found in parts of the middle temporal gyrus (BA22) and the middle temporal gyrus (BA39). (**B**) A shifted transverse and coronal view of the right hemisphere and the occipital region, showing cortical activations on the three-dimensionally rendered Colin27 template [89]. A, anterior; P, posterior; L, Left; R, Right. MNI, Montreal Neurological Institute; X, Y, Z, corresponding MNI coordinates; BA, Brodmann area.

**Figure 7 brainsci-07-00076-f007:**
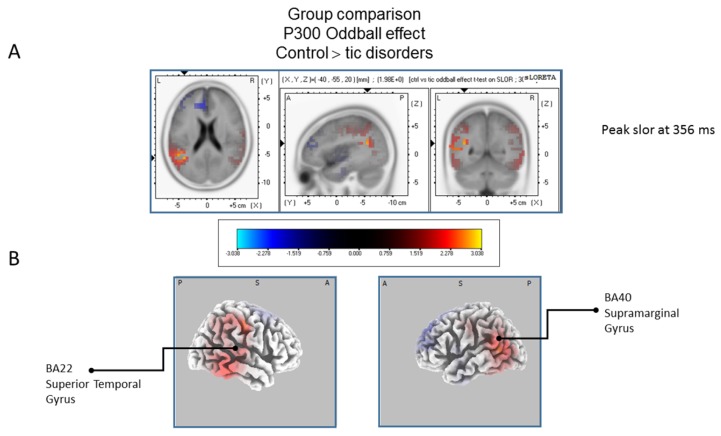
Results of the tic disorder patients based on the standardized low-resolution brain electrotomography (sLORETA) current source density localization of the oddball effect (contrast:“Ctrl” > “Tic”) in the mean activation of the P300 component within a 300–550 ms time window. Images have been obtained after statistical non-parametric mapping (SnPM) and co-registration to the stereotaxic Talairach space based on the Co-Planar Stereotaxic Atlas of the Human Brain [83] and the probabilistic MNI-152 template [81]. There is no significant activated voxels (after correction for multiple comparisons (*p* < 0.05)). (**A**) Although non-significant, the peak of highest cortical activity has been found in parts of the temporal gyrus (BA22), supramarginal gyrus (BA40). (**B**) A shifted transverse and sagittal view of the left and the right hemisphere, showing cortical activations on the three-dimensionally rendered Colin27 template [89]. A, anterior; P, posterior; MNI, Montreal Neurological Institute; X, Y, Z, corresponding MNI coordinates; BA, Brodmann area.

**Table 1 brainsci-07-00076-t001:** Sociodemographic and clinical characteristics. Means (standard deviations).

	Tic Disorders (A) (*n* = 12)	BFRB Disorders (B) (*n* = 12)	Control (C) (*n* = 15)	ANOVA ^1^
Age (years old)	33 (9)	30 (8)	31 (9)	ns (*p* = 0.71; ŋ^2^ < 0.001)
**Sex (M/F)**	**7/5**	**1/11**	**5/10**	**A vs. B (*p* = 0.04; ŋ^2^ = 0.28)**
Schooling (years)	14 (2)	15 (2)	14 (2)	ns (*p* = 0.37; ŋ^2^ < 0.07)
Non-verbal intelligence (Raven Progressive Matrices)	90 (9)	74 (18)	81 (18)	ns (*p* = 0.06; ŋ^2^ = 0.15)
**Anxiety (BAI)**	**6 (6)**	**11 (8)**	**5 (5)**	**B vs. C (*p* = 0.03; ŋ^2^ = 0.18)**
Depression (BDI)	11 (12)	10 (5)	4 (4)	ns (*p* = 0.08; ŋ^2^ = 0.13)
OCS (VOCI)	25 (21)	24 (15)	-	ns (*p* = 0.90; Cohen’s d = 0.06)
Tic severity (YGTSS)	35 (16)	25 (7)	-	ns (*p* = 0.06; Cohen’s d = 0.82)
BFRB severity (MGH-HPS)	-	14 (3)	-	-
BFRB severity (MGH-adapted for trichotillomania)	-	13 (6)	-	-
BFRB severity (MGH-adapted for onycophagia)	-	12 (5)	-	-
BFRB severity (MGH-adapted for skin-picking)	-	12 (5)	-	-
BFRB severity (MGH-adapted for skin-scratching)	-	15 (5)	-	-

^1^ Kruskall–Wallis test was used for non-parametric data (sex); BAI: Beck Anxiety Inventory (max. score 63); BDI: Beck Depression Inventory (max. score 63); BFRB: Body-focused repetitive behaviors; MGH-HPS: Massachusetts General Hospital-Hair Pulling Scale; ns: non-significant VOCI: Vancouver Obsessional and Compulsive Inventory (max. score 220); OCS: Obsessive-Compulsive Symptoms; YGTSS: Yale Global Tic Severity Scale (max. score 100). Significant results are in bold.

**Table 2 brainsci-07-00076-t002:** Standardized low-resolution brain electromagnetic tomography (sLORETA) results of maximal brain electrical activity for the oddball effect (rare vs. frequent: 518 ms latency) for the TD group.

Brain Regions	BA	Hemisphere	Lobe	Coordinates (X, Y, Z)						*t*-Values		Effect Sizes
				MNI			Talairach MNI			Max Voxel Value	No of Activated Voxels	
Precentral/Medial/Middle Frontal/Sub gyral	6	L/R	Frontal	−20	−5	70	−20	−2	65	3.72 *	291	0.75
Cingulate Gyrus	24	L	Limbic	−15	−5	50	−15	−3	46	3.03 *	91	0.68
Paracentral Lobule	31	R	Limbic	10	−15	50	10	−12	47	2.90 *	35	0.66
Superior Frontal Gyrus	10	L	Frontal	−30	50	30	−30	50	25	2.63 *	53	0.62
Superior and medial Frontal Gyrus	9	L	Frontal	−10	50	35	−10	50	30	2.58 *	73	0.62
Medial Frontal Gyrus/Cingulate Gyrus	32	L	Frontal	−5	5	50	−5	7	46	2.57 *	19	0.61
Superior Occipital Gyrus/Cuneus/Precuneus	19	R	Occipital	35	−85	30	35	−81	32	2.41 *	105	0.59
Precentral Gyrus	4	R	Frontal	45	−15	55	45	−12	51	2.41 *	24	0.59
Postcentral Gyrus	3	R	Parietal	50	−15	55	50	−12	51	2.37 *	11	0.58
Angular Gyrus	39	R	Temporal	35	−80	30	35	−76	31	2.33 *	3	0.58
Paracentral Lobule	5	M	Frontal	0	−30	55	0	−27	52	2.32 *	7	0.57
Cingulate Gyrus	23	M	Limbic	0	−15	35	0	−13	33	2.32 *	11	0.57
Superior Frontal Gyrus	8	L	Frontal	−20	45	45	−20	46	39	2.25 *	7	0.56
Cuneus	18	R	Occipital	20	−85	25	20	−81	27	2.25 *	20	0.56
Somatosensory association cortex	7	R	Occipital	15	−80	30	15	−76	31	2.19 *	31	0.55

Talairach/MNI coordinates and *t*-values correspond to the maximum peak activity in each brain region. * *p* < 0.05; L: left; R: right; M: midline; max: maximum; BA: Brodmann area; MNI: Montreal Neurological Institute.

**Table 3 brainsci-07-00076-t003:** Standardized low-resolution brain electromagnetic tomography (sLORETA) results of maximal brain electrical activity for the oddball effect (rare vs. frequent: 402 ms latency) for the BFRB group.

Brain Regions	BA	Hemisphere	Lobe	Coordinates (X, Y, Z)						*t*-Values		Effect Sizes
				MNI			Talairach MNI			Max Voxel Values	No of Activated Voxels	
Postcentral Gyrus/Precentral Gyrus	43	R	Parietal	65	−15	20	64	−14	19	4.13 **	12	0.78
Inferior Parietal/Postcentral Gyrus	40	R	Parietal	65	−25	25	64	−23	24	4.11 **	121	0.78
Postcentral Gyrus	3	R	Parietal	65	−15	25	64	−13	24	4.05 **	25	0.77
Transverse Temporal/Superior Temporal	42	R	Temporal	60	−10	15	59	−9	14	4.02 **	20	0.77
Postcentral Gyrus	1	R	Parietal	65	−20	30	64	−18	29	3.97 **	7	0.77
Precentral Gyrus	4	R	Frontal	60	−10	25	59	−9	23	3.97 **	21	0.77
Postcentral Gyrus	2	R	Parietal	60	−20	30	59	−18	29	3.92 **	40	0.76
Precentral Gyrus	6	R	Frontal	50	−5	20	50	−4	19	3.88 **	42	0.76
Superior Temporal	22	R	Temporal	65	−5	15	64	−4	14	3.85 **	55	0.76
Inferior Frontal Gyrus	44	R	Frontal	50	0	20	50	1	18	3.84 **	29	0.76
Inferior Frontal Gyrus	9	R	Frontal	50	0	25	50	1	23	3.76 **	51	0.75
Transverse Temporal Gyrus	41	R	Temporal	55	−20	10	54	−19	10	3.55 **	15	0.73
Inferior Frontal Gyrus	45	R	Frontal	60	10	20	59	11	18	3.54 **	33	0.73
Insula	13	R	Sub-lobar	40	0	20	40	1	18	3.53 **	66	0.73

Talairach/MNI coordinates and *t*-values correspond to the maximum peak activity in each brain region. ** *p* < 0.005 * *p* < 0.05; L: left; R: right; N: number; min: minimum; max: maximum; BA: Brodmann area; MNI: Montreal Neurological Institute.

**Table 4 brainsci-07-00076-t004:** Standardized low-resolution brain electromagnetic tomography (sLORETA) results of maximal brain electrical activity for the oddball effect (rare vs. frequent: 348 ms latency) for the control group.

Brain Regions	BA	Hemisphere	Lobe	Coordinates (X, Y, Z)						*t*-Values		Effect Sizes
				MNI			Talairach MNI			Max Voxel Values	No of Activated Voxels	
Cuneus	18	R	Occipital	15	−80	25	15	−76	27	2.36 **	40	0.53
Somatosensory association cortex	7	R	Occipital/Parietal	15	−80	30	15	−76	31	2.34 **	44	0.53
Precuneus	19	R	Occipital	25	−90	30	25	−86	32	2.33 **	76	0.53
Precuneus	31	R	Occipital/Parietal	10	−75	25	10	−72	27	2.32 **	20	0.53
Postcentral Gyrus	3	R	Parietal	45	−20	50	45	−17	47	2.09 **	4	0.49
Precentral Gyrus	4	R	Parietal	45	−20	45	45	−17	42	2.05 **	2	0.48
Middle Temporal Gyrus	21	R	Temporal	60	−15	−10	59	−15	-8	2.06 **	5	0.47

Talairach/MNI coordinates and *t*-values correspond to the maximum peak activity in each brain region. ** *p* < 0.005; L: left; R: right; max: maximum; BA: Brodmann area; MNI: Montreal Neurological Institute.

**Table 5 brainsci-07-00076-t005:** Standardized low-resolution brain electromagnetic tomography (sLORETA) results of maximal brain electrical activity for the Control vs BFRB group difference (at 356 ms latency).

Brain Regions	BA	Hemisphere	Lobe	Coordinates (X, Y, Z)						*t*-Values		Effect Sizes
				MNI			Talairach MNI			Max Voxel Value	No of Activated Voxels	
Superior Frontal Gyrus	6	R	Frontal	20	−10	70	20	−6	65	−4.81 **	128	0.69
Middle Temporal Gyrus	39	L	Temporal	−50	−75	20	−50	−72	22	−3.53 *	27	0.57
Middle Frontal Gyrus	8	R	Frontal	30	20	45	30	21	40	−3.28 *	32	0.54
Middle Occipital Gyrus	19	L	Occipital	−55	−75	5	−54	−72	8	−3.27 *	42	0.54
Middle Temporal Gyrus	22	L	Temporal	−40	−60	15	−40	−57	17	−3.27 *	2	0.54
Middle Occipital Gyrus	37	L	Occipital	−55	−75	0	−54	−73	4	−3.16 *	15	0.53
Superior Frontal Gyrus	10	L	Frontal	−10	60	30	−10	60	25	3.13 *	8	0.52

Talairach/MNI coordinates and *t*-values correspond to the peak activity in each brain region. ** *p* < 0.005; * *p* < 0.05; L: left; R: right; max: maximum; BA: Brodmann area; MNI: Montreal Neurological Institute.

**Table 6 brainsci-07-00076-t006:** Standardized low-resolution brain electromagnetic tomography (sLORETA) results of maximal brain electrical activity for the TD vs. BFRB group difference (at 324 ms latency).

Brain Regions	BA	Hemisphere	Lobe	Coordinates X, Y, Z						*t*-Values		Effect Sizes
				MNI			Talairach			Max Voxel Value	No of Activated Voxels	
Middle Temporal Gyrus	22	R	Temporal	40	−60	15	40	−57	17	−5.56 **	19	0.75
Middle Temporal Gyrus	39	R	Temporal	45	−65	20	45	−62	22	−5.34 **	36	0.74
Fusiform Gyrus	37	R	Temporal	35	−50	−15	35	−49	-10	−4.86 **	85	0.70
Parahippocampal Gyrus	19	R	Limbic	30	−50	−5	30	−49	−2	−4.81 **	59	0.70
Inferior Temporal Gyrus	20	R	Temporal	50	−50	−20	50	−49	−14	−4.39 **	13	0.67
Supramarginal Gyrus	40	R	Temporal	60	−55	20	59	−52	21	−4.37 **	7	0.67
Parahippocampal Gyrus	30	R	Limbic	25	−55	0	25	−53	3	−4.29 **	3	0.66
Middle Temporal Gyrus	21	R	Temporal	60	−60	0	59	−58	3	−4.10 **	10	0.64

Talairach/MNI coordinates and *t*-values correspond to the peak activity in each brain region. ** *p* < 0.005; L: left; R: right; N: number; min: minimum; max: maximum; BA: Brodmann area; MNI: Montreal Neurological Institute.

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
