# Peer review of "P300 Source Localization Contrasts in Body-Focused Repetitive Behaviors and Tic Disorders"

_brainsci, 2017, doi:10.3390/brainsci7070076_

Round 1

Reviewer 1 Report

The authors carried out an interesting study with the aim of developing a reliable method of discriminating between Tic Disorder (TD) and body-focused repetitive behaviours (BFRB). For this purpose, they recorded ERPs elicited by a motor oddball task in three groups of participants: TD, BFRB and controls. The careful selection of participants with TD and with BFRB is of note.

The authors found that the P300 amplitude in response to infrequent stimuli in the right hemisphere was significantly lower in the participants with BFRB than in the other two groups of participants. The authors also use sLORETA to describe the pattern of cortical activation associated with the oddball effect in each of the three groups.

The part of the Discussion concerning the cortical activation of the oddball effect is rather weak. The authors assume that there are differences between groups in this activation, on the basis of the data processed by sLORETA for each of the three groups, without having applied statistical tests to compare the groups. The authors must carry out statistical comparisons of the three groups with the sLORETA-derived data and then rewrite the Discussion on the basis of the results obtained. 

Minor points:

In the Statistical Analysis subsection Electrode site factor (5 levels) the authors should specify what the 5 levels are, indicating the active electrodes comprising each of these levels. 

In Results, in the subsection  Amplitude of  the P300 component, lines 281, 282 and 283, each of the 5 levels of the Electrode site factor are named and the three electrodes forming them are indicated. These names do not seem appropriate considering the active electrodes referred to. The authors should choose more intuitively suitable names that would correspond to the active electrodes.

Author Response

RESPONSE LETTER

REVIEWER #1

The authors carried out an interesting study with the aim of developing a reliable method of discriminating between Tic Disorder (TD) and body-focused repetitive behaviours (BFRB). For this purpose, they recorded ERPs elicited by a motor oddball task in three groups of participants: TD, BFRB and controls. The careful selection of participants with TD and with BFRB is of note.

The authors found that the P300 amplitude in response to infrequent stimuli in the right hemisphere was significantly lower in the participants with BFRB than in the other two groups of participants. The authors also use sLORETA to describe the pattern of cortical activation associated with the oddball effect in each of the three groups.

The part of the Discussion concerning the cortical activation of the oddball effect is rather weak. The authors assume that there are differences between groups in this activation, on the basis of the data processed by sLORETA for each of the three groups, without having applied statistical tests to compare the groups. The authors must carry out statistical comparisons of the three groups with the sLORETA-derived data and then rewrite the Discussion on the basis of the results obtained.

RESPONSE: The current set of LORETA analysis was guided by the results of the repeated measures ANOVA that revealed a Group x Condition by Hemisphere by Electrode interaction. This interaction was decomposed by condition and showed that a Hemisphere by Electrode by Group and a Region by Hemisphere by Electrode by Group interactions were present only in the Rare condition.  Because the topography and the condition effects were significant for certain groups, we focused on separate analyses per condition in order to contrast the topographies of the condition effect (oddball effect) in each group.  However, to comply with the comment, we carried out additional statistical comparisons of the three groups with the sLORETA-derived data. The same procedure was done to display differences to compare the three group differences on the P300 time oddball effect (e.g., “tic disorders” > “control”; “tic disorders” > “BFRB” ; “BFRB” > “control”).

These analyses confirmed the ANOVA results and showed that the oddball effect is different between Tic and BFRB and also between control and BFRB, mainly around the right hemisphere. There is no difference between the control participants and the tic disorders group. A paragraph was added in the result section line 335-348 and also through lines 248-251. Three figures were added (figure 5,6,7) as well as table 5 and 6 for the significant effects only.

Minor points:

In the Statistical Analysis subsection Electrode site factor (5 levels)- the authors should specify what the 5 levels are, indicating the active electrodes comprising each of these levels.

RESPONSE: We thank the reviewer for this comment of precision. We have specified the active electrodes comprising each level and have also indicated the active electrodes for the midline analysis, in which the Electrode site factor comprised 6 levels.

The text now reads as follow line 220-227: “One between-subject factor with three levels (Group: TD, BFRB, Control) was applied and 4 within-subject factors that comprise: Condition (Rare, frequent), Region (Frontal, central, parietal), Hemisphere (Left, right) and Electrode sites (5 levels as follow: Left Frontal = AF3, AF1, F5, F3, F1; Right Frontal = AF2, AF4, F2, F4, F6; Left Central = FC3, FC1, C5, C3, C1; Right Central = FC2, FC4, C2, C4, C6; Left Parietal = CP5, CP1, P5, P3, P1; Right Parietal = CP2, CP6, P2, P4, P6). Additionally, analyses were conducted with midline electrodes with the same between-subject factor and the following within-subject factors: Condition (Rare, Frequent) and Electrode sites (6 levels; AFZ, FZ, FCZ, CZ, CPZ, PZ).”

In the Result section, in the subsection - Amplitude of the P300 component, lines 281, 282 and 283, each of the 5 levels of the Electrode site factor are named and the three electrodes forming them are indicated. These names do not seem appropriate considering the active electrodes referred to. The authors should choose more intuitively suitable names that would correspond to the active electrodes.

RESPONSE: We decided to simply specify the electrodes in the names to be as precise and clear as possible.

The text now reads as follow: “The latter interaction of Electrode by Group was decomposed according to the following electrode lines (i.e. 1st AF4/FC4/CP6; 2nd AF2/FC2/CP2; 3rd F6/C6/P6; 4th F4/C4/P4; 5th F2/C2/P2).”

Reviewer 2 Report

Review of “P300 source localization contrasts in body-focused repetitive behaviors and tic disorders”

The present paper is a well-written comparison of a neural marker (i.e., P300) of working memory in individuals with tic disorders, body-focused repetitive behaviors, and healthy controls using a motor oddball task. A significant strength is that the research group was able to exclude individuals on psychiatric medications and with clinically significant comorbidity which helps with drawing conclusions regarding the findings. BFRB and tic symptoms are not less often compared in research studies so this work is interesting and novel.

Introduction:

The introduction thoroughly describes the prior literature on this topic. I wonder, are there any studies showing comorbidity rates between BFRBs and tic disorders? It seems like both BFRBs and tic disorders are more comorbid with OCD than one another. However, historically, existing structured diagnostic interviews have not allowed for good assessment of comorbid BFRBs in studies.

Measures:

I suggest adding more information on the structure of the measures, especially since the YGTSS was adapted to be used for BFRBs.

Method:

No issues

Discussion:

The authors provided a good discussion of their work in the context of the extant literature. The mention of the OCD in the discussion is highly relevant. With respect to limitations, the authors do mention the small sample size as a limitation in the highlights section above acknowledgments, but should also mention this in the main discussion text.

It is worth highlighting the heterogeneity of the BFRB group either as a limitation or elsewhere as a potential confound. The authors somewhat allude to this in the last few sentences of the discussion when they mention the potential benefits of future inclusion of an OCD group. However, it could be stated more explicitly.

Tables:

I suggest reporting the actual p-values and effect sizes in the tables; and report the actual p-values in the text unless they are lower than .001.  

Author Response

RESPONSE LETTER

REVIEWER #2

Review of “P300 source localization contrasts in body-focused repetitive behaviors and tic disorders”

The present paper is a well-written comparison of a neural marker (i.e., P300) of working memory in individuals with tic disorders, body-focused repetitive behaviors, and healthy controls using a motor oddball task. A significant strength is that the research group was able to exclude individuals on psychiatric medications and with clinically significant comorbidity which helps with drawing conclusions regarding the findings. BFRB and tic symptoms are not less often compared in research studies so this work is interesting and novel.

Introduction:

The introduction thoroughly describes the prior literature on this topic. I wonder, are there any studies showing comorbidity rates between BFRBs and tic disorders? It seems like both BFRBs and tic disorders are more comorbid with OCD than one another. However, historically, existing structured diagnostic interviews have not allowed for good assessment of comorbid BFRBs in studies.

RESPONSE: Indeed, that point is crucial. While there are no studies assessing comorbidity between tic disorders and BFRB as a whole, few studies reported comorbidity between TD and either trichotillomania or skin-picking. We added a few sentences to reflect this and to explain how our study could fill the gap in the literature: “To our knowledge, no study has reported the comorbidity rates between tic disorders and BFRB as a whole. Yet, few studies reported such rates for individual BFRB. For instance, trichotillomania was reported in about 3% of TD patients [16,17], but could occur more frequently in TD+OCD patients [18]. In a sample of 35 TD patients, Ghanizadeh and Mosallaei [19] reported nail-biting in almost 30% of them. This comorbidity between TD and BFRB highlights the need for objective assessment tools to distinguish them.”

Measures:

I suggest adding more information on the structure of the measures, especially since the YGTSS was adapted to be used for BFRBs.

RESPONSE: We added some precisions about the TSGS and the YGTSS: “The TSGS combines a clinical evaluation with reports made by patients and parents to assess tics (simple/complex and motor/phonic) and social functioning in the past week. Global scores vary from 0 (no symptoms) to 100 (worst Tourette possible). Inter-rater reliability, construct validity, and concurrent validity were all found to be good [68]. The YGTSS is a symptom checklist for tic severity (number of tics, frequency, intensity, complexity and interference) and functioning impairments (distress experienced in interpersonal, academic and occupational spheres) over the previous week. The YGTSS global score (0-100) is the sum of subscales evaluating the severity of tics (0-50) and daily impairment (0-50). The YGTSS has demonstrated good internal consistency, test-retest stability, inter-rater agreement and convergent validity (with the TSGS) [69-71].”

Method:

No issues

Discussion:

The authors provided a good discussion of their work in the context of the extant literature. The mention of the OCD in the discussion is highly relevant. With respect to limitations, the authors do mention the small sample size as a limitation in the highlights section above acknowledgments, but should also mention this in the main discussion text.

RESPONSE: A sentence regarding the small sample size was added in the main discussion text: “While our study has some limitations, as the small sample size, the exclusion of medicated and comorbid TS patients represents a great strength.”

It is worth highlighting the heterogeneity of the BFRB group either as a limitation or elsewhere as a potential confound. The authors somewhat allude to this in the last few sentences of the discussion when they mention the potential benefits of future inclusion of an OCD group. However, it could be stated more explicitly.

RESPONSE: We added a precision regarding the heterogeneous nature of BFRB: “Although they are all classified in the same category and broadly conceptualized as BFRB, their clinical presentation can be rather heterogeneous, which could confound our results.”

Tables:

I suggest reporting the actual p-values and effect sizes in the tables; and report the actual p-values in the text unless they are lower than .001.  

RESPONSE: We thank the reviewer for this comment and have made the changes accordingly throughout the manuscript and the tables.

Round 2

Reviewer 1 Report

The authors have now included the statistical analyses required. I think that the manuscript should be published once the authors correct the following:

Results

Page 5, line 224: "5 levels" is wrong, are 6 levels

Page 9, line 398: "frequent-rare subtraction", it is best rare-frequent

Page 9, line 402: "Figure 5" does not match, it is figure 6.

Page 9, line 406: "figure 6" does not match, it is figure 5

Page 9, line 406: authors have to describe the differences between groups

Discussion

Page 21, line 602-604: Only meaningful results must be discussed. Since the authors found no significant difference between the control group and the TD group, lines 602, 603 and 604 should be eliminated. The authors should be very careful about what they write, because what appears on line 4 is not true, they found no significant differences between controls and with TD in the upper frontal gyrus.

Author Response

The authors have now included the statistical analyses required. I think that the manuscript should be published once the authors correct the following:

Results

Page 5, line 224: "5 levels" is wrong, are 6 levels

RESPONSE: We thank the reviewer for this comment and have made the appropriate correction.

Page 9, line 398: "frequent-rare subtraction", it is best rare-frequent

RESPONSE: We are grateful to the reviewer for this suggestion and have made the change.

Page 9, line 402: "Figure 5" does not match, it is figure 6.

RESPONSE: We thank the reviewer for this comment. We changed the order of the figures so that the text would remain unchanged.

Page 9, line 406: "figure 6" does not match, it is figure 5

RESPONSE: We changed the order of the figures so that the text would remain unchanged.

Page 9, line 406: authors have to describe the differences between groups

RESPONSE: We are thankful to the reviewer for this suggestion and the modified text now reads as follow:

“The table 6 showed that, in the tic/BFRB comparison, a significantly larger difference in cortical activation of the oddball effect peaked at 324 ms post-stimulus significant (p’s between 0.0004 to 0.0040) in the right hemisphere middle temporal gyrus (BA22) and middle temporal gyrus (BA39) (see figure 6) confirming that BFRB patients showed a significantly reduced P300 oddball effect compared to the TD patients, mainly over the right middle temporal gyrus.”

Discussion

Page 21, line 602-604: Only meaningful results must be discussed. Since the authors found no significant difference between the control group and the TD group, lines 602, 603 and 604 should be eliminated. The authors should be very careful about what they write, because what appears on line 4 is not true, they found no significant differences between controls and with TD in the upper frontal gyrus.

RESPONSE: We thank the reviewer for this suggestion and have removed the problematic sentence.
